# In Vivo Study of Organ and Tissue Stability According to the Types of Bioresorbable Bone Screws

**DOI:** 10.3390/ma17225632

**Published:** 2024-11-18

**Authors:** Tae-Young Kwon, Geum-Hwa Lee, Hyuk Lee, Kwang-Bok Lee

**Affiliations:** 1Department of Orthopedics Surgery, Jeonbuk National University Hospital, Jeonju 54907, Republic of Korea; malaise@naver.com (T.-Y.K.); dlgur6750@gmail.com (H.L.); 2Research Institute of Clinical Medicine, Biomedical Research Institute, Jeonbuk National University, Jeonbuk National University Hospital, Jeonju 54907, Republic of Korea; heloin@jbnu.ac.kr

**Keywords:** magnesium alloy, polylactic acid screw, in vivo study, biocompatibility

## Abstract

Biodegradable material, such as magnesium alloy or polylactic acid (PLA), is a promising candidate for orthopedic surgery. The alloying of metals and the addition of rare earths to increase mechanical strength are still questionable in terms of biosafety as absorbent materials. Therefore, the purpose of this study is to understand the effect of substances due to the degradation of various biodegradable substances on organs in the body or surrounding tissues. A total of eighty male Sprague−Dawley rats were selected for this study, and the animals were divided into four groups. Each of the three experimental groups was implanted with magnesium alloy, polymer, and titanium implants; the control group only drilled into the cortical bone. Serum assay, micro-CT, hematoxylin and eosin staining, immunoblotting, and real-time PCR were evaluated. There was no significant difference between the two groups of magnesium alloy and polymer in serum assay, but micro-CT analysis confirmed that magnesium alloy degrades faster than polymer, and histological examination showed a strong inflammatory response in the early stages, which was similarly observed in immunoblotting and real-time PCR. Our findings show that there was no toxicity due to the degradation of the biodegradable material, and the difference in each inflammatory response is thought to be determined by the rate of degradation in the body.

## 1. Introduction

In the field of orthopedic surgery, the use of metallic implants such as titanium plates, screws, and pins has been commonly utilized to stabilize fractured bones and to facilitate the healing process. These implants are renowned for their exceptional mechanical strength and reliability in fixing bone fractures, which have established them as the gold standard for fracture fixation and orthopedic reconstruction [1,2]. Despite their effectiveness in providing robust mechanical support, and thereby contributing successful patient outcomes in clinical settings, these metallic implants usually require secondary surgical procedures for removal once the bone has sufficiently healed. This subsequent surgery not only increases the overall cost of treatment but also introduces additional risks and potential complications, such as surgical site infections, longer recovery periods, and a greater burden on healthcare resources [3].

To mitigate these issues, there is an ongoing quest to identify alternative materials that could obviate the need for such invasive procedures. In this pursuit, there has been growing interest in the development and application of biodegradable materials, such as magnesium alloys and polylactic acid (PLA) [4,5,6]. These bioresorbable materials are designed to naturally degrade within the body, thereby eliminating the need for a secondary surgical intervention. However, despite these advantages, the materials present their own set of challenges, particularly in terms of mechanical strength and fixation stability compared to traditional metallic implants [7,8,9]. Efforts to enhance the performance of biodegradable materials include alloying with specific ions (e.g., Zn, Ca) and incorporating rare earth elements to improve mechanical properties [10]. Nevertheless, concerns remain regarding the biocompatibility and safety of these materials as they degrade and interact with biological tissues.

The primary objective of this study is to comprehensively identify the effects of degradation substances from various biodegradable materials on organs or surrounding tissues in the body. The degradation process of these materials can result in the release of by-products that can interact with biological tissues and systemic organs [10]. These degradation products may affect local and systemic environments, leading to potential adverse effects on surrounding tissues and vital organs. Understanding these interactions is crucial for assessing the overall safety and efficacy of biodegradable implants.

This study aims to address critical gaps in in the knowledge surrounding biodegradable implants by comprehensively investigating the impact of degradation products from magnesium alloy and PLA implants [6,11]. Using a controlled animal model, this study will focus on their impact on peri-implant tissues as well as on vital organs such as the liver and kidneys to ensure that these materials do not pose significant health risks. Additionally, this study will employ a range of evaluation methods, including serum assays, micro-CT imaging, histological examination, and molecular analyses to provide a detailed understanding of the safety and performance of these biodegradable materials.

For this research, we have chosen a bone implantation model to facilitate detailed examination of peri-implant tissue responses. This model allows for an in-depth analysis of tissue reactions at the implant site, systemic toxicity, and overall impact on organ function. By employing this approach, we aim to provide valuable insights into the safety and efficacy of magnesium alloy and PLA implants, ultimately contributing to the optimization and advancement of biodegradable orthopedic technologies.

## 2. Materials and Methods

### 2.1. Materials Preparation

For the magnesium alloy implant, cortex Mg−5Ca screws (HP Mg, more than 99.98 wt.%; 5 wt.% Ca) with an outer diameter of 2.0 mm and a length of 6.0 mm were used (Resomet^TM^; U&I Corporation, Seoul, Republic of Korea). With the polylactic acid (PLA)-based implant, an outer diameter of 2.7 mm and a length of 40 mm were used (FreedonScrew^TM^; INION OY, Tampere, Finland). For the titanium implant, a diameter of 2.0 mm and a length of 6.0 mm were used (TDM Corporation, Gwangju, Republic of Korea).

### 2.2. Animals and Surgery

The current study was conducted in compliance with the Declaration of Helsinki and was approved by the Institutional Animal Care and Use Committee of the Jeonbuk National University Laboratory Animal Center (JBUH-IACUC-2021-44).

A total of eighty male Sprague−Dawley rats (7 weeks old), weighing 200−250 g, were selected for this study. All animals were kept under identical conditions, maintaining a constant temperature of 25 ± 2 °C, a humidity of 55% ± 5%, and a standard light–dark cycle, with unrestricted access to food and water throughout the research.

For the in vivo experiment, the animals were divided into four groups, each consisting of 20 animals based on the type of samples administered. Each of the three experimental groups was implanted with magnesium alloy, polymer, and titanium implants, one control group only drilled into the cortical bone. Typically, sterilization methods for medical implants include autoclaving, ethylene oxide gas, or gamma irradiation. Previous studies have shown that ethylene oxide gas provides a stable sterilization method for magnesium alloy and PLA, so this method was selected [12,13].

The rats were anesthetized via intraperitoneal injection of 0.6 mL/kg tiletamine and zolazepam (Zoletil 50, Virbac Laboratories, Carros, France) and 0.4 mL/kg xylazine hydrochloride (Rompun, Bayer Korea, Seoul, Republic of Korea).

The surgical area was shaved and disinfected with an iodine scrub, and a 1 cm incision was made over the femur. The periosteum was then elevated by performing a full-layer incision with an #11-blade, and using a periosteal elevator the periosteum was gently dissected from the bone. The implant placement was 1 cm above the knee joint line on the lateral femoral condyle. The implants were manually screwed after drilling a hole (Figure 1). A total of 5 mice per group were sacrificed for micro-CT and histological assessment in the 1st, 2nd, 4th, and 8th weeks, post-surgery, by administering an overdose of ketamine (100 mg/kg). The body weight and weight of liver and kidney tissues was measured.

### 2.3. Serum Assay of Biochemical Parameter

The rat was fasted for 15 h prior to euthanasia. Blood samples were collected from a truncal vein and centrifuged at 1100 × g for 15 min at 4 °C to separate serum and stored at −70 °C. The serum levels of aspartate aminotransferase (AST), alanine aminotransferase (ALT), lactate dehydrogenase (LDH), and creatinine were analyzed using an automatic biochemical analyzer (Hitachi-7020, Hitachi Medical, Tokyo, Japan).

### 2.4. Assessment of Bone Microstructure Using Micro-Computed Tomography

The microstructure of bone was evaluated using a micro-computed tomography (CT) scanner. The fourth lumbar vertebrae were scanned using the X-ray μCT SkyScan 1076 (SkyScan, Aartselaar, Belgium) system at a resolution of 35 μm, as outlined in prior research. Three-dimensional images were generated and analyzed with Ant software (version 2.4; Bruker, Kontich, Belgium).

### 2.5. Immunoblotting and Real-Time PCR Analysis

Immunoblotting was performed as described previously [14]. Briefly, muscle tissue samples were sonicated, electrophoresed on a polyacrylamide gel, and subsequently transferred to PVDF membranes. After blocking, these membranes were incubated with primary antibodies for IL-6, TNF-α, IL-1β, and β-actin (Santa Cruz Biotechnologies). Following this, membranes were washed and treated with species-specific horseradish peroxidase-conjugated secondary antibodies. The protein bands were visualized using an ECL system (Bio-Rad, Hercules, CA, USA) and quantified through the Image J software version 1.54 (NIH, Bethesda, MD, USA).

A real-time polymerase chain reaction assay (RT-PCR) was performed as described previously [15]. Briefly, RNA was extracted from homogenized muscle tissue using TRIzol (Invitrogen), and 2 μg of RNA was utilized for cDNA synthesis with oligo dT primers and reverse transcriptase. Real-time PCR was conducted using the ABI 7500 system (Applied Biosystems, Foster City, CA, USA) with the TaKaRa SYBR Premix Ex Taq Kit (TaKaRa Bio Inc., Tokyo, Japan). The expression levels of the genes were determined using the comparative cycle threshold (Ct) method. Specific primers used are detailed in Table 1.

### 2.6. Hematoxylin and Eosin Staining

Post-euthanasia, tibia bone samples were fixed in 4% formalin for 48 h. The samples then underwent a month-long decalcification process in an EDTA solution, with the solution being refreshed every three days. Cross-sections of 5 μm were obtained from the embedded samples and stained with hematoxylin and eosin for histological examination.

### 2.7. Statistical Analysis

SPSS software version 25.0 (IBM, Armonk, NY, USA) was used to conduct a statistical analysis of the data. Comparisons between three groups were evaluated using two-way ANOVA using Tukey’s post-hoc test. Statistical significance was confirmed when the *p*-value was <0.05.

## 3. Results

### 3.1. Systemic Toxicity Evaluation

Body, tissue weight, and serum biochemical parameters are listed in Table 2. In the first week, other parameters did not show statistical significance between each group, but LDH, ALT, and AST levels in the Ti group were significantly lower compared to the other groups (*p* < 0.05). In the second week, creatinine levels in the Mg and Ti groups showed statistically significant increases compared to other groups (*p* < 0.05). In the fourth week, LDH levels in the sham and Ti groups showed statistically significant increases compared to other groups (*p* < 0.05). In the eighth week, creatinine and ALT in the Ti group were significantly lower than those in the other groups (Figure 2).

### 3.2. Micro-CT Evaluation

Representative micro 3D CT results obtained from the femur–implant complex are shown in Figure 3. As a result of a CT follow-up, all implants were in place in the first week of the experiment. From the second week, the magnesium alloy screws were regraded, and a gas cavity could be observed around them. In the fourth week, new bone formation can be seen. However, due to the PLA screw’s relatively slow degradation characteristic, it maintained its state until the eighth week. In the sham group, bone healing of the drilled hole was observed from the second week of surgery. This study did not measure the amount of gas formation identified in previous studies, nor the amount or rate of degradation of the magnesium alloy screw.

### 3.3. Analysis of Protein and Gene Expression

To confirm the tissue inflammatory response in each group, the inflammatory cytokines TNF-α, IL-1b, and IL-6 were examined by immunoblot analysis. Compared to the control group, the magnesium alloy screw showed a significant increase of about three times until the fourth week (*p* < 0.05) and then decreased from the eighth week. In the polylactide group, it peaked at the second week and then showed a decline pattern. In the Ti group, TNF and IL showed no significant difference from the control group, and the IL-6 increased from the fourth week (Figure 4).

### 3.4. Histologic Analysis for the Surrounding Bone Tissue by H and E Staining

Tissue reaction to the implant was characterized by the infiltration of inflammatory leukocytes (macrophages, lymphocytes, and neutrophils). Groups were divided and scored according to the number of infiltrated cells. In the first week, there was no significant difference in the amount of inflammatory cell infiltration between the inserted screws, but in the second and fourth weeks the magnesium alloy group had more infiltration of inflammatory cells than the other groups, showing a statistically significant difference. This shows similar results to the results of immunoblotting and real-time PCR analysis (Figure 5).

## 4. Discussion

The objective of this in vivo study was to evaluate the short-term organ and tissue stability of two different types of biodegradable implant systems using a rat model. Magnesium alloy or PLA implants may leave debris during degradation leading to local inflammatory responses and foreign-body reactions [16,17]. There were several studies about organ and tissue stability of each magnesium alloy and polymer implant, but in this study, we compared the differences between both biodegradable implants.

It is known that each implant has the biodegradable property, but the degradation mechanism and the speed are different. PGA, which was the first generation of resorbable systems, showed degradation that was too rapid [18]. PLA, the next generation system, provided sufficient healing time before resorption. It is generally believed that the mechanical properties of PLA depend on its molecular weight, and the high molecular weight of PLA possesses high intermolecular forces, which can induce high mechanical properties [19,20]. However, the high molecular weight will lead to a slower degradation rate in a human body due to its hydrophobicity and high crystallinity, and the complete degradation takes 2~3 years [21,22]. PLA can be degraded by many mediums, such as water, sunlight, microorganisms, and enzymes. In a physiological environment, PLA is mainly degraded under the action of water molecules, and the degradation process can be divided into two stages: (1) The ester bond of the PLA chain breaks and water molecules begin to penetrate the PLA matrix, causing the molecular weight to decrease and producing water-soluble oligomers. (2) The oligomers are gradually hydrolyzed into monomeric lactic acids under the catalysis of biological enzymes, and then further metabolized into water and carbon dioxide [23].

In a physiological environment, Mg is mainly degraded under the action of water molecules. The different gas formation rates and dissolution rates of Mg ions can be explained as follows [24]:Mg + 2H_2_O → Mg (OH)_2_ + H_2_

Gas formation initially occurs when Mg and H_2_O in the body fluid come into contact [25].

To evaluate the safety of the Mg and PLA implants, we measured body, liver and kidney weight. We found for all implant groups that the body, liver, and kidney weight had no significant differences compared to the control and sham groups. This showed similar values to the results of weight changes measured in previous magnesium alloy studies [26].

As another method to confirm biocompatibility, serum analysis was performed, and LDH was analyzed to determine muscle toxicity. Creatine was nephrotoxic, and ALT and AST were analyzed to determine hepatotoxicity [27]. Creatine, AST, and ALT increased in all groups until the second week, and then decreased and were elevated within normal ranges. In other studies, similarly, liver and kidney functions in the blood were found to be not affected by the degradation of magnesium alloys after implantation into the body, confirming the general safety of degradable Mg alloys [26,27,28]. However, in the case of LDH, it showed a decreasing trend at the fourth week and increased at the eighth week. The increase in LDH without an increase in liver and kidney enzymes at week eight is expected to be due to muscle damage, but in the case of histological findings, immunoblotting, and real-time PCR, inflammatory findings initially increased and then decreased. In this regard, it appears that follow-up research is needed to determine whether there are other factors.

Biodegradable materials implanted in the body have different degradation rates depending on the implanted bone area, soft tissue, and body fluid composition. In addition, there are differences depending on the surface treatment method and alloy or polymer composition [23,29,30]. In general, it is known that the decomposition of magnesium alloy screws implanted in the bone shaft is faster than in the epiphysis [31]. In addition, magnesium alloy implants are known to have a faster degradation rate than polymer implants.

As a result of a CT follow-up, all implants were in place in the first week of the experiment, but from the second week, the magnesium alloy screws were regraded, and a gas cavity could be observed around them. At the fourth week, new bone formation could be seen. Still, the complications of hydrogen gas produced from magnesium alloy implants remain controversial. Some preclinical studies reported gas formation during degradation without specific complications [32,33]. Other in vivo studies indicated that the rapid and persistent hydrogen gas cavity formation led to prolonged discomfort and affected blood cell formation, which decreased the survival rate of rats [34]. However, due to the PLA screw’s relatively slow degradation characteristic, it maintained until the eighth week. In the sham group, bone healing of the drilled hole was observed from the second week of surgery.

The degree of local inflammatory response due to the degradation of biodegradable screws and the expression of inflammatory cytokines was analyzed through western blotting and PCR. The expression pattern of inflammatory cytokines was similar to the decomposition rate pattern of magnesium alloy and polymer. Magnesium alloy showed rapid degradation up to the first four weeks, and the expression of inflammatory cytokines also increased up to four weeks and then decreased thereafter. In the case of polymers, it is believed that the initially increased inflammatory cytokines were expressed due to tissue damage following surgery, not due to polymer decomposition.

Histological examination showed proliferation of inflammatory cells in the magnesium alloy group until the fourth week, but it gradually decreased thereafter and normal tissue findings were observed. Our results are similar to other animal experiments with the implantation of Mg with a screw and without a significant inflammatory reaction with the surrounding bone [35]. In the case of PLA, there is a possibility of chronic inflammation or foreign-body reaction due to its relatively slow degradation rate [17,36,37].

Mg-based alloys have properties such as higher strength-to-weight ratios, good machineability, and castability. Mg-based implants have higher yield strength with a low Young’s modulus that is close to the natural bone [38]. However, pure Mg alloys are highly degradable in the biological environment. To overcome this problem, addition of alloying elements such as calcium, aluminum, lithium, and different rare earth materials aid in controlling the degradation rate of Mg and improving the mechanical properties [39]. In our study, to control degradation rate, Mg-5wt%Ca was used. In a recent in vivo study of a Mg-5wt%Ca alloy screw, the degree of biodegradation of magnesium in the body is known to be proportional to gas formation, showing an increase up to 12 weeks and then a decrease. During this period, most fractures were observed to have healed [33].

Another bioabsorbable material, the polymer implant, also has advantages such as good compatibility, high chemical resistance, low density, and good elasticity. However, compared to Mg alloy or Ti materials, they have the disadvantage of relatively weaker mechanical strength. A combination of two or more polymers can greatly enhance the properties and overcome the shortcomings of polymer implants [40]. INION^TM^ bioresorbable implants made of Trimethylene Carbonate TMC, L-Lactide LPLA and D, and L-Lactide DLPLA degrade in vivo by hydrolysis, subsequently being metabolized by the organism in CO_2_ and H_2_O. Degradation profiles were adapted to provide initial stability and then progressive bone loading to stimulate regeneration. The implants gradually lost their resistance after 18–36 weeks in vivo with complete resorption between 2 and 4 years [41].

There are several potential limitations to this study with regard to interpretation of the data presented. First, the bioabsorbable implant was not sufficiently degraded due to the short research period of about eight weeks. Additionally, the degradation rate of each implant was not determined using micro-CT during this period. These are matters that need to be further experimented with in subsequent studies.

## 5. Conclusions

In this study, serum analysis, immunoblotting, and tissue histologic examination were performed to determine the biocompatibility of magnesium alloy and PLA, which are currently commonly used in biodegradable screws. In the early period, a mild inflammatory reaction was observed, but it was not severe enough to cause problems with biocompatibility. Therefore, both implants suggest potential as a promising candidate for medical applications. It is considered that it is important to understand the differences in the decomposition speed of each implant and apply them to clinical practice in an appropriate manner. Although numerous biodegradable implants have been developed, they have not yet succeeded in fully replacing conventional implants. Nevertheless, this study aims to contribute to a future increase in the adoption of biodegradable implants.

## Figures and Tables

**Figure 1 materials-17-05632-f001:**
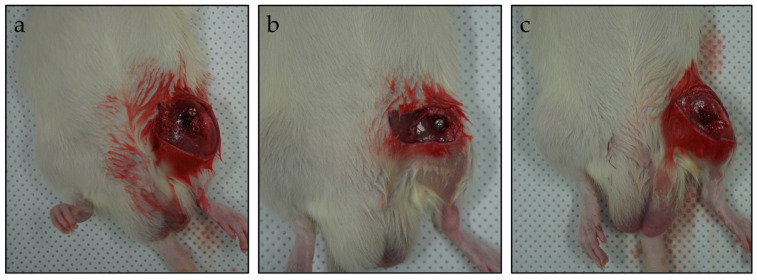
Implant placement at distal femur, (**a**) magnesium alloy screw, (**b**) titanium screw, (**c**) PLA screw.

**Figure 2 materials-17-05632-f002:**
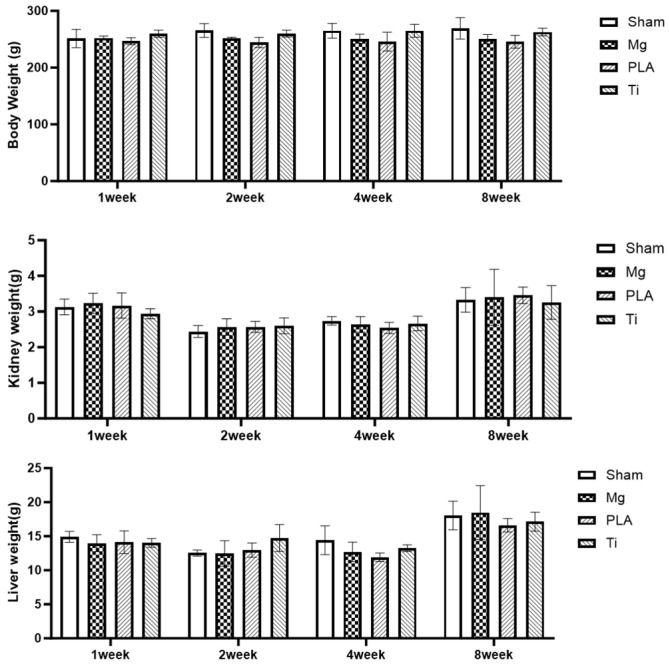
Body, kidney, and liver weight records and serum LDH, Creatine, AST, and ALT measurements. Data represents the arithmetic mean ± standard deviation (SD). * *p* < 0.05, ** *p* < 0.001.

**Figure 3 materials-17-05632-f003:**
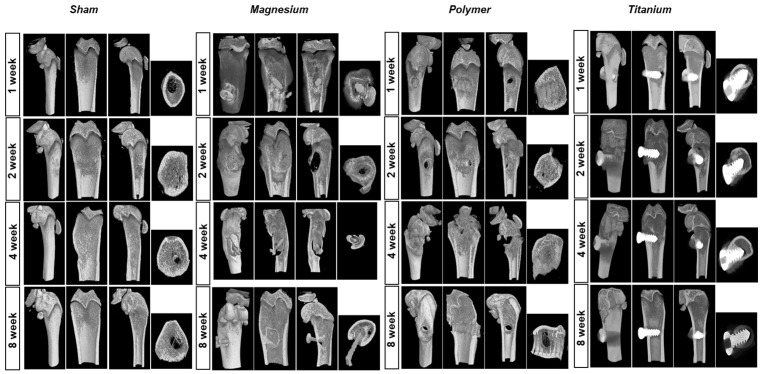
Micro-CT images of implants placed in distal femur. In the magnesium alloy screw group, gas cavity formation happens at the second week and new bone formation happens at the fourth week.

**Figure 4 materials-17-05632-f004:**
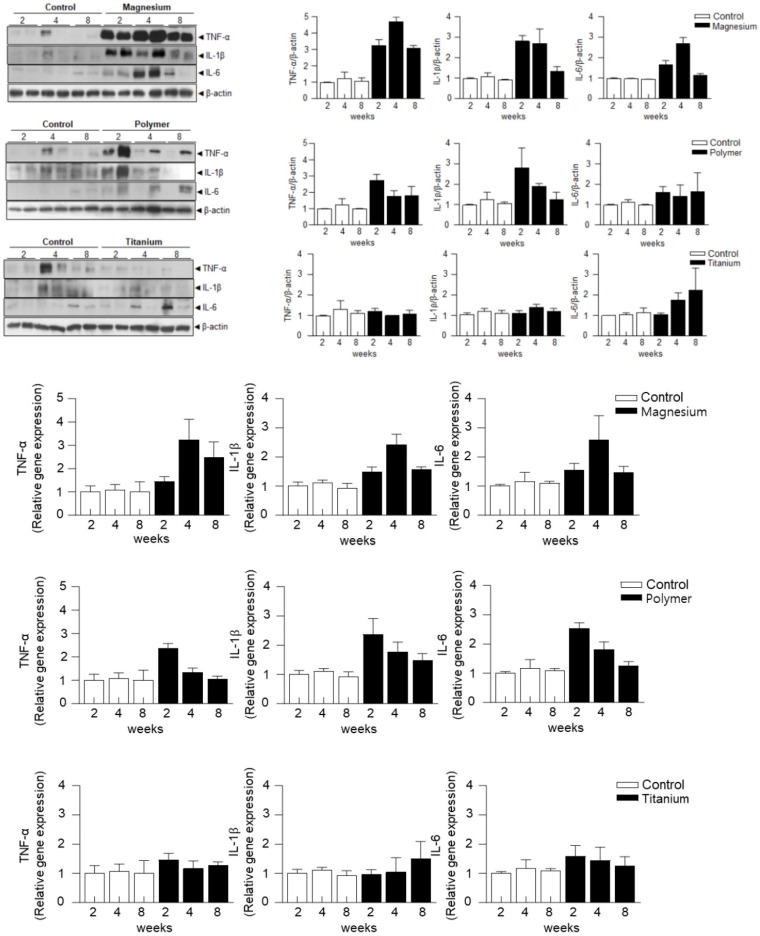
Western blotting and RT-PCR analysis of muscle tissue samples. The expression of TNF-α, IL-1b, and IL-6 was significantly higher until the fourth week, and then decreased from the eighth week.

**Figure 5 materials-17-05632-f005:**
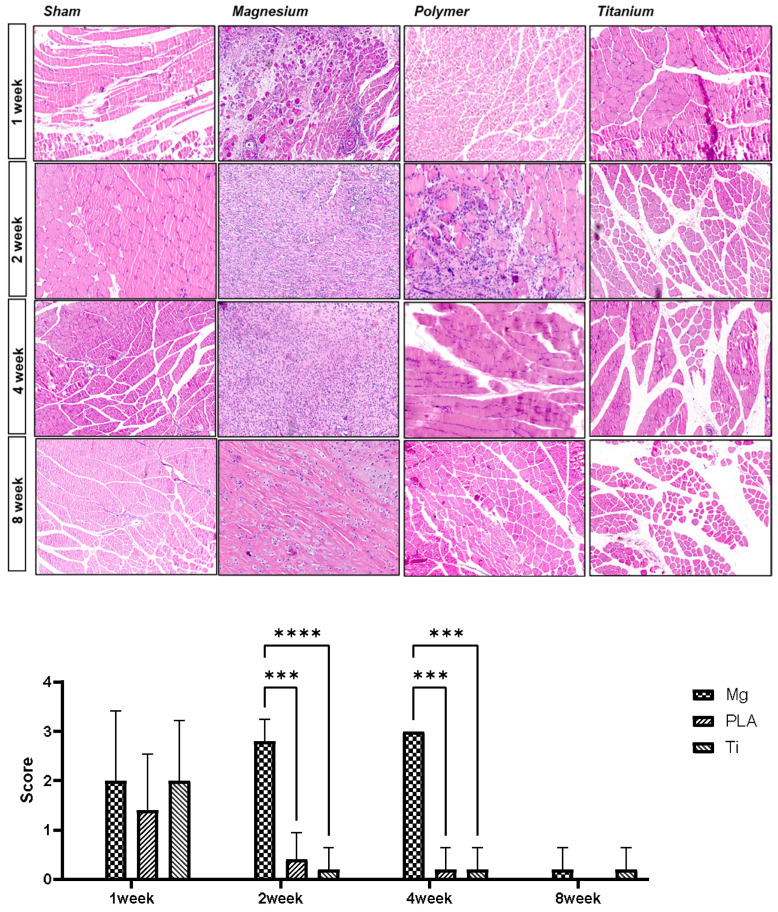
Hematoxylin-eosin staining and tissue inflammation score (***, ****, *p* < 0.05).

**Table 1 materials-17-05632-t001:** List of primers for real-time qPCR analysis.

Gene	Forward Primer (5′–3′)	Reverse Primer (5′–3′)
TNF-α	CAGGGGCCACCACGCTCTTC	CTTGGGGCAGGGGCTCTTGA
IL-6	AGGGCATGTTAAGGAGC	CATCAGAGGCAAGGAGGA
IL-1β	GCAACTGTTCCTGAACTCAACT	ATCTTTTGGGGT CCGTCAACT
β-actin	TTCAACACCCCAGCCATGT	CAGTGGTACGACCAGAGGCATA

**Table 2 materials-17-05632-t002:** Average weight of body, liver, kidney tissues and serum assays.

Group	Implantation Time	Body Weight (g)	Kidney Weight (g)	Liver Weight (g)	LDH (U/L)	Creatine (mg/dl)	ALT (U/L)	AST (U/L)
Sham	1 week	251.3 ± 16.1	3.13 ± 0.22	14.92 ± 0.79	742.2 ± 180.58	0.37 ± 0.04	21.8 ± 5.23	51.9 ± 5.71
	2 weeks	265.5 ± 12.2	2.43 ± 0.17	12.52 ± 0.45	996.00 ± 112.59	0.53 ± 0.03	38.08 ± 5.98	73.64 ± 11.34
	4 weeks	265.1 ± 13.1	2.74 ± 0.12	14.42 ± 2.12	764.40 ± 229.34	0.46 ± 0.05	34.86 ± 7.71	60.92 ± 7.37
	8 weeks	269.4 ± 18.8	3.32 ± 0.35	18.03 ± 2.11	1632.40 ± 274.03	0.48 ± 0.03	26.54 ± 5.27	81.10 ± 10.17
Magnesium Alloy	1 week	252.2 ± 2.5	3.23 ± 0.28	13.97 ± 1.23	709.80 ± 69.35	0.41 ± 0.04	21.00 ± 2.18	61.04 ± 4.41
	2 weeks	252.1 ± 1.4	2.56 ± 0.23	12.48 ± 1.87	1061.40 ± 194.65	0.61 ± 0.03	42.00 ± 8.00	83.08 ± 14.79
	4 weeks	251.1 ± 8.2	2.64 ± 0.22	12.69 ± 1.43	286.20 ± 175.76	0.44 ± 0.05	27.70 ± 5.15	47.68 ± 9.19
	8 weeks	250.1 ± 8.4	3.39 ± 0.79	18.48 ± 3.98	1707.80 ± 672.28	0.53 ± 0.03	27.38 ± 3.24	87.44 ± 16.51
Polylactide	1 week	246.7 ± 6.1	3.17 ± 0.35	14.11 ± 1.68	1107.80 ± 177.48	0.40 ± 0.03	21.70 ± 1.83	70.94 ± 12.11
	2 weeks	244.5 ± 8.9	2.56 ± 0.15	12.95 ± 1.05	902.80 ± 104.98	0.53 ± 0.05	35.50 ± 5.48	69.04 ± 5.87
	4 weeks	245.9 ± 16.2	2.54 ± 0.18	11.80 ± 0.69	165.75 ± 46.76	0.45 ± 0.05	31.10 ± 8.87	44.20 ± 8.63
	8 weeks	244.5 ± 11.3	3.45 ± 0.23	16.60 ± 0.99	1010.60 ± 333.49	0.51 ± 0.06	25.62 ± 4.36	66.14 ± 14.52
Ti	1 week	259.7 ± 6.3	2.93 ± 0.14	14.00 ± 0.65	403.40 ± 150.95	0.39 ± 0.04	17.68 ± 2.07	45.82 ± 6.40
	2 weeks	259.9 ± 6.1	2.60 ± 0.22	14.73 ± 1.97	713.20 ± 278.45	0.64 ± 0.03	37.38 ± 2.69	68.58 ± 10.63
	4 weeks	265.1 ± 11.3	2.66 ± 0.47	13.23 ± 0.51	771.20 ± 207.11	0.46 ± 0.05	34.70 ± 12.18	58.32 ± 11.99
	8 weeks	262.9 ± 6.7	3.25 ± 0.45	17.13 ± 1.39	1010.60 ± 333.49	0.43 ± 0.02	20.42 ± 1.47	67.54 ± 10.45

## Data Availability

Data are contained within the article.

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
