# Peer review of "In Vivo Study of Organ and Tissue Stability According to the Types of Bioresorbable Bone Screws"

_materials, 2024, doi:10.3390/ma17225632_

Round 1
Reviewer 1 Report
Comments and Suggestions for Authors
This manuscript concerns an in vivo study of two types of bioresorbable bone screws. Comparison between magnesium alloy and polylactic-acid has been carried out using total eighty male Sprague−Dawley rats, in which they were divided into four groups (implanted with magnesium, polymer, or titanium implants, and one control group only drilled into the cortical bone). Based on the findings of this study, the authors confirmed that both implants (magnesium alloy and polylactic-acid) are suggesting the potential as a promising candidate for medical applications. In my opinion, the idea of this manuscript is good, and the experiments were technically well performed.
This paper may be considered a welcome addition to the current literature, which attempts to understand the bioresorbability behavior of implants and apply them to clinical practice in an appropriate manner. However, this paper contains some limitations that are not discussed or even mentioned and should be clarified, especially with regard to sterilization and discussion.
Comments:
1. Animals and Surgery: Were the implants sterilized to avoid any contamination before applying them? The authors should specify the sterilization method used for the implants. Can they explain the choice of this method?
2. Discussion: A comparative discussion should be proposed according to existing bioresorbable bone screws.
Minors:
- Abstract: The term 'magnesium’ should be written as 'magnesium alloy’.
- Conclusions: The authors should highlight the implications and perspectives of the study.
Author Response
1. Animals and Surgery: Were the implants sterilized to avoid any contamination before applying them? The authors should specify the sterilization method used for the implants. Can they explain the choice of this method?
Thank you for pointing this out. We revised manuscript highlighted.
2. Discussion: A comparative discussion should be proposed according to existing bioresorbable bone screws.
Thank you for pointing this out. We revised manuscript highlighted.
Minors:
- Abstract: The term 'magnesium’ should be written as 'magnesium alloy’.
Thank you for pointing this out. We revised manuscript highlighted.
- Conclusions: The authors should highlight the implications and perspectives of the study.
Thank you for pointing this out. We revised manuscript highlighted.
Reviewer 2 Report
Comments and Suggestions for Authors
This manuscript presents an in vivo study of organ and tissue stability using two types of bioresorbable bone screws. The study evaluates the biocompatibility and degradation of magnesium (Mg) and polylactic acid (PLA) screws compared to titanium (Ti) screws in a rat model. Magnesium degraded faster than PLA, leading to early inflammatory responses and gas cavity formation by the second week, while new bone formation was observed by the fourth week. In contrast, PLA degraded more slowly, with minimal inflammatory response. Despite the early inflammation observed in the magnesium group, both materials were found to be biocompatible, showing no significant toxicity to vital organs like the liver and kidneys. Overall, magnesium and PLA screws are promising biodegradable implants, with differences in degradation rates being critical for clinical application. This study is within the scope of the journal Materials and addresses an important gap in the field of biodegradable orthopedic implants. The references are appropriate. The methodology appears adequate, but there is one significant issue:
-Mechanical analysis is necessary. The study focuses on biocompatibility but does not assess the mechanical properties or fixation stability of the implants over time. Mechanical testing of bone-implant integration would provide valuable insights into the implants' structural integrity during degradation. This is a well-known concern in the literature, as previous studies have highlighted the importance of mechanical stability in biodegradable implants. (Here I could accept also explanation related to previous literature explaining this problem.)
Minor issues:
-The study should address expectations from longer-duration studies, as the 8-week study period may not be sufficient to fully evaluate the long-term degradation and biocompatibility of PLA, which degrades slowly. Previous literature indicates that PLA can take years to fully degrade, and this limitation should be discussed.
-In section 2.4, the description of the micro-CT system "Skyscann 1076 system" is not detailed enough. It should be written as: "X-ray μCT SkyScan 1076 (SkyScan, Aartselaar, Belgium)" in accordance with the instruction manual of the instrument.
-The conclusions section needs significant improvement, as it currently lacks depth and detail. A more comprehensive summary of the findings and implications for future research and clinical application is necessary.
Author Response
-Mechanical analysis is necessary. The study focuses on biocompatibility but does not assess the mechanical properties or fixation stability of the implants over time. Mechanical testing of bone-implant integration would provide valuable insights into the implants' structural integrity during degradation. This is a well-known concern in the literature, as previous studies have highlighted the importance of mechanical stability in biodegradable implants. (Here I could accept also explanation related to previous literature explaining this problem.)
Thank you for pointing this out. We revised manuscript highlighted.
Minor issues:
-The study should address expectations from longer-duration studies, as the 8-week study period may not be sufficient to fully evaluate the long-term degradation and biocompatibility of PLA, which degrades slowly. Previous literature indicates that PLA can take years to fully degrade, and this limitation should be discussed.
Thank you for pointing this out. We revised manuscript highlighted.
-In section 2.4, the description of the micro-CT system "Skyscann 1076 system" is not detailed enough. It should be written as: "X-ray μCT SkyScan 1076 (SkyScan, Aartselaar, Belgium)" in accordance with the instruction manual of the instrument.
Thank you for pointing this out. We revised manuscript highlighted.
-The conclusions section needs significant improvement, as it currently lacks depth and detail. A more comprehensive summary of the findings and implications for future research and clinical application is necessary.
Thank you for pointing this out. We revised manuscript highlighted.
Round 2
Reviewer 1 Report
Comments and Suggestions for Authors
No further comments. The authors addressed all my review's concerns.
Reviewer 2 Report
Comments and Suggestions for Authors
The manuscript is ready to be accepted for publication as all issues have been addressed.